# Extracting alignment data in open models

**Federico Barbero** [1]   **Xiangming Gu** [2 3]   **Christopher A. Choquette-Choo** [1 4]   **Chawin Sitawarin** [1]
**Matthew Jagielski** [5 3]   **Itay Yona** [6 3]   **Petar Veličković** [1]   **Ilia Shumailov** [7 3]   **Jamie Hayes** [1]

## Abstract

In this work, we show that it is possible to extract significant amounts of *alignment training data* from a post-trained model – useful to steer the model to improve certain capabilities such as long-context reasoning, safety, instruction following, and maths. While the majority of related work on memorisation has focused on measuring success of training data extraction through string matching, we argue that embedding models are better suited for our specific goals. Distances measured through a high quality embedding model can identify semantic similarities between strings that a different metric such as edit distance will struggle to capture. In fact, in our investigation, approximate string matching would have severely undercounted (by a conservative estimate of $10\times$) the amount of data that can be extracted due to trivial artifacts that deflate the metric. Interestingly, we find that models readily regurgitate training data that was used in post-training phases such as SFT or RL. We show that this data can be then used to train a base model, recovering a meaningful amount of the original performance. We believe our work exposes a possibly overlooked risk towards extracting alignment data. Finally, our work opens up an interesting discussion on the downstream effects of *distillation* practices: since models seem to be regurgitating aspects of their training set, distillation can therefore be thought of as indirectly training on the model's original dataset.

## 1. Introduction

[1]Google DeepMind [2]National University of Singapore [3]Work performed while at Google DeepMind [4]Now at OpenAI [5]Anthropic [6]MentaLeap [7]AI Sequrity Company. Correspondence to: Federico Barbero <fedz@google.com>.

*Proceedings of the $43^{rd}$ International Conference on Machine Learning*, Seoul, South Korea. PMLR 306, 2026. Copyright 2026 by the author(s).

[0]Here we use term 'training' data loosely and elaborate on it later in the Nomenclature section in the Appendix (Section A).

Progress in capabilities of Large Language Models (LLMs) is frequently driven by improvements to training data recipes. It is common for a model developer to curate targeted data bundles to push performance on particular downstream benchmarks. For the purpose of this work, we refer to this data as '*alignment*' data. We use the term to encompass not only data used for safety and instruction-following (such as Supervised Finetuning (SFT) and Reinforcement Learning (RL) datasets), but also any targeted data collections used to steer model behaviour and enhance specific capabilities, including mathematics, reasoning, and long-context understanding. While this type of data is usually found in post-training, it is becoming common to include it also earlier in training (Meta, 2025). We use the term alignment rather than post-training in our work for this reason.

The fact that models memorise subsets of their training data is now a well-established phenomenon (Carlini et al., 2021; 2022; Nasr et al., 2025; Feldman, 2020; Biderman et al., 2023; Kiyomaru et al., 2024; Kandpal et al., 2022; Brown et al., 2021; Borec et al., 2024; Chen et al., 2024a; Stoehr et al., 2024; Haviv et al., 2022; Leybzon & Kervadec, 2024; Jagielski et al., 2022; Mireshghallah et al., 2022; Prashanth et al., 2024; Schwarzschild et al., 2024; Zhang et al., 2023; Lu et al., 2024; Hartmann et al., 2023; Hayes et al., 2025b;a; Morris et al., 2025). Most research on this topic has focused on this issue due to its associated privacy and legal risks, such as models leaking personal or copyrighted information (Freeman et al., 2024; Cooper et al., 2025; Cooper & Grimmelmann, 2024; Mueller et al., 2024; Karamolegkou et al., 2023; Ippolito et al., 2022; Satvaty et al., 2024; Lee et al., 2023; Huang et al., 2022). Prior work on memorisation is often centred around verbatim or near-verbatim training data extraction (Huang et al., 2024; Carlini et al., 2021; 2022), where success is measured by exact (or very close) matches on tasks where this is important, like measuring similarity of a credit card number or a paragraph from a copyrighted book. In contrast, in this work we study and develop a more subtle notion of training data extraction – patterns and templates of proprietary data – where semantic structure is just as valuable as the literal content. Consequently, existing extraction methods and metrics used to determine failure or success using simple string matching are not well-aligned for this task.

We are interested in understanding if an LLM will leak training data that is sensitive due to its utility in improving model performance. In other words, if a model's competitive advantage comes from its secret training data, and models have a tendency to memorise, and then regurgitate this data, then the competitive advantage itself may be at risk. This is especially topical with the surge of more sophisticated training modes such as the new *thinking* paradigm (Nye et al., 2021; Wei et al., 2022; Reynolds & McDonell, 2021). It is also important from the point of view of the commonplace practice of *model distillation*, where a competitor may use a strong model to train their own. If models regurgitate training data, then through distillation a competitor is (at least in part) training on the original training data as well.

By exploiting the fact that in open-weight models, the end user controls tokenization, and that the chat template structure is only introduced during post-training, we use a simple attack which demonstrates open-weight models will repeat numerous training prompts [1]. We are able to extract data ranging from reinforcement learning (RL) prompts and associated traces to prompts used for supervised fine-tuning (SFT) and during mid and pre-training. In doing so, we answer an open question from Xu et al. (2024) and confirm that models do indeed recite their own alignment data and that therefore distillation pipelines are likely to partly include original training data samples.

---

### Summary

**Our hypothesis:** Since the chat template is exclusively introduced in post-training, it can be used to generate alignment data.

**Our procedure:**

1. Prompt the model with the chat template and sample. Repeat this a number of times to generate a set of synthetic data points.

2. For each synthetic data point, find the closest sample in the post-training dataset according the embedding similarity, using an embedding model.

**Our main finding:** One may reasonably expect the synthetic data points to be from the same distribution as the alignment dataset. We find that this is true, but it is far closer to the alignment dataset that one may expect. For example, in Figure 1, we generated a maths question and answer that is near identical to a sample in the alignment dataset.

---

[1]Our attack relies on the user's control over the chat template structure. This is not normally available for closed models, and so our attack does not immediately apply to them.

**Contributions**    We outline our contributions and findings:

- We generalise the prompting strategy from Xu et al. (2024) and study how it may be used to extract post-training data from models trained with SFT and RL, addressing their open question on memorisation.

- We show that measuring the quantity of memorisation based on approximate or verbatim string matching is likely significantly undercounting rates of memorisation – our results suggesting at least by $10\times$. Instead, using neural embeddings reveals a much higher rate.

- We finally demonstrate that our method can be used to generate datasets from a model that can then be used for the (post-)training of another model, meaningfully capturing some of the original model's performance.

## 2. Extracting alignment data with neural embeddings

The proposed extraction strategy is based on the observation that certain prompts seem to consistently induce the model into outputting alignment-like data. To enable our attack, we use special tokens from the chat template that are precisely introduced during post-training, making them ideal artifacts that can be leveraged to extract specific types of data. *Our main contribution* is confirming that many of such generations are either verbatim or very close to true training samples under an appropriate measure of similarity.

Our pipeline works as follows: we embed the entire post-training set using an embedding model (see 2.3 for details), constructing a vector search engine. We then generate a number of samples simply by prompting the model using our chosen prefix repeatedly. For each generated sample we then embed it and search the vector database to retrieve the best match and its score. A diagram of this process is shown in Figure 1. Once the post-training dataset is embedded, the search can be implemented as a single matrix multiplication. In the following Sections 2.1 and 2.2, we will provide motivation for our methodology using generated examples.

A significant constraint of our work is that we require access to models that make their (post-)training mixtures available. For this reason, we focus our study on OLMo 2 (OLMo et al., 2024) (Section 3) for SFT and Open-Reasoner-Zero (Hu et al., 2025) (Section 4) for RL. As these models are post-trained with standard methods, they are a valuable resource for our study.

### 2.1. Chat templates to steer generations

The observation that chat templates can be used to generate useful synthetic data has been pointed out already by (Xu

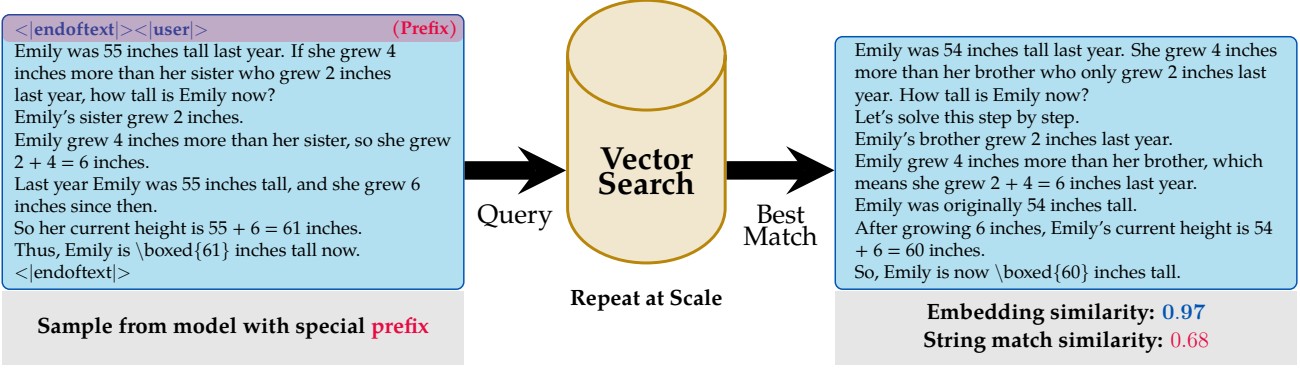

*Figure 1.* An overview of the data extraction process. A sample generated using the prompt <|endoftext|><|user|>\n is used as a query for a vector search to find the best semantic match from the training data. The figure highlights the difference between a high embedding similarity score (0.97) and a much lower string-match similarity score (0.68), demonstrating that semantic similarity is more effective for detecting this form of data memorisation.

et al., 2024), where the authors use special prompts and a filtering pipeline to generate a dataset that can be used to post-train models using SFT. In our work, we study this from a different angle and aim to understand the extent to which the generations correspond to regurgitated training data. In particular, we positively answer one of the conjectures left from (Xu et al., 2024), which posited that the data generated might have been training data.

In Figure 2, we show how the prompting the model with the chat template results in outputs that resemble post-training samples. We will investigate this phenomenon in much more detail in the coming sections.

### 2.2. Measuring aproximate semantic memorisation

A central theme in our work is the wish to broaden the definition of memorisation beyond simple string matching, due to its natural limitations. For example, string matching has been shown to be vulnerable to 'style-transfer' (Ippolito et al., 2022), where semantics are preserved whilst easily evading string matching checks. Further, we are interested in the usefulness of extracted samples as training points; if two samples are semantically equivalent, then they reasonably should be treated equal as training samples (for measurement of memorisation rate). While approximate string matching may (accidentally) weakly capture semantics in certain cases, we found it to be unreliable and generally not well-aligned with the notion of semantic equivalence.

We use Figure 3 as a small illustrative example from a generation which came from our pipeline. It is clear that the left and right samples are extremely similar up to small differences, for example in the numerical contents of the choices in the multiple choice section of the question. String-matching scores tend to penalise these differences quite heavily assigning a similarity of $\approx 0.7$, while the high embedding score (see Section 2.3 for details) arguably

---

**Qwen2.5 7B Instruct**

<|im_start|>user
I'm trying to use the 'zip' function in Python with a generator expression, but I keep encountering a 'ValueError: too many values to unpack'. Here's the code I'm working with:

```python
gen = (x for x in range(3))
a, b = zip(*gen)
```

**[...]**

---

**Qwen2.5 7B Instruct**

<|im_start|>user
Okay, so I need to figure out the value of the expression $\sqrt{2 + \sqrt{2 + \sqrt{2 + \sqrt{2 + \cdots}}}}$. Hmm, let me think. I remember that infinite nested radicals can sometimes be solved by setting them equal to a variable and then solving an equation. Let me try that approach here.

Let me denote the entire expression as $x$. So, $x = \sqrt{2 + \sqrt{2 \ldots}}$
**[...]**

*Figure 2.* Generations from Qwen2.5 7B Instruct using <|im_start|>user\n as a prefill. Qualitatively, the generations resemble potential post-training samples.

aligns better with the memorisation judgement a human would make. We observed many situations where a string-matching score would provide low similarity while the samples are semantically extremely similar. We point out that

under the standard 0.9 approximate memorisation threshold (Comanici et al., 2025), the example in Figure 3 would count as *not* memorised. We delay to Section 3.1 a more detailed investigation of the limitations of string matching.

---

**Generation**

<|endoftext|><|user|>
The question is: Find the value of y from
`(12)^3 x 6^4 ÷ 432 = y?`
Options:
(A) 2456
(B) 4578
(C) 6234
(D) 7386
(E) 5184
The answer is: (E)
Explanation: Given Exp. = (12)3 x 64 = (12)3 x 64 = (12)2 x 62 = (72)2 = 5184
432 12 x 64<|endoftext|>

---

**SFT Training sample**

Question: Find the value of y from
`(12)^3 x 6^4 ÷ 432 = y?`
Options:
(A) 2356
(B) 3467
(C) 4567
(D) 5189
(E) 5184

Random thoughts:
Given Exp. = (12)3 x 64 = (12)3 x 64 = (12)2 x 62 = (72)2 = 5184
432 12 x 62 The final answer: (E).

---

*Figure 3.* (Left) Generation from OLMo 2 13B. (Right) True post-training sample. Neural embeddings match provide a score of 0.986 using `gemini-embedding-001` (Lee et al., 2025), while normalised Levenshtein similarity provides a match score of 0.699, heavily penalising differences in the options, even though the semantics remain identical. When computing similarities we always strip out special tokens, highlighted in blue. We report in the Appendix (Section E) more of such examples.

## 2.3. Measuring memorisation with neural embeddings

Embedding scores are generated by an embedding model and can be tuned for different tasks. The process usually involves taking a strong base model and training it contrastively for a specific application (e.g., semantic retrieval) (Lee et al., 2024). We use the `gemini-embedding-001` model (Lee et al., 2025) in all of our experiments as it is a general and strong embed-

ding model. We generate a single embedding for each sample, *removing all special tokens* and therefore only considering the plain text. This acts as a vector search engine, where we can compute similarity with respect to each training sample using a single matrix multiplication and taking the `argmax`. As embeddings are normalised to be unit-norm, their dot product naturally encodes a notion of similarity. To better distinguish this from approximate memorisation measures using string matching, we call this a measure using embeddings a measure of **approximate semantic memorisation**.

To define a sample as (approximately semantic) memorised, we need to choose an appropriate threshold. We do this manually and report randomly chosen samples at different thresholds in the Appendix (Sections E and F). We chose a threshold of 0.95 for neural embeddings as a more conservative choice. We found samples at this point having qualitatively similar properties that would be indisputable given a reasonable human judge. The choice of the threshold will naturally affect the measured memorisation. This limitation is however also present in choosing a threshold for what is considered memorised according to a string matching metric.

## 3. Large scale extraction of SFT data

We focus our SFT memorisation study on OLMo 2 (OLMo et al., 2024)[2]. OLMo 2 comes with the full training mixture, alongside the training details. Further, the models have strong downstream performance, enabling us to conduct experiments that result in findings that are likely generalizable to other models. The uncompressed pre-training mix has a size of 22.4 TB while the higher quality mid-training split has a size of 5.14 TB. The post-training is divided in 3 stages, the first is an SFT stage with a dataset containing 939k samples, then a Direct Preference Optimisation (Rafailov et al., 2023) step is conducted with 378k samples, and finally a RL with Verifiable Rewards (RLVR) step with 29.9k samples. We focus on the extraction of the SFT data in this section.

We apply the procedure we described in Section 2 using neural embeddings. We embed the SFT training samples using `gemini-embedding-001` by concatenating the question and answer sequences as a single block of text. To extract the data from the model, we generate conditioning on the following initial tokens <|endoftext|><|user|>\n, which are the starting tokens of the chat template. We leave the temperature at the default value of 1.

## 3.1. String matching poorly captures memorisation

We start by evaluating the memorisation using traditional string matching metrics. We consider 100k generations

---

[2]Licensed by AllenAI under the Apache License, Version 2.0.

for OLMo 2 13B (OLMo et al., 2024) using our extraction method and search for their closest match in the post-training set, with respect to different similarity measures. We consider the normalised Levenshtein similarity defined as $1 - \texttt{Levenshtein}(A, B)/\texttt{max(len(A),len(B))}$ and the normalised Indel similarity defined as $1 - \texttt{Indel}(A, B)/(\texttt{len(A)+len(B)})$. The Indel similarity is related to the Levenshtein distance, but applies a cost of 2 to substitutions. For each generated sample, we find the highest similarity based on the two string matching methods in the post-training set. We follow the heuristic used by Gemini 2.5 (Comanici et al., 2025) and characterise a sample as *approximately memorised* when its similarity is above 0.9.

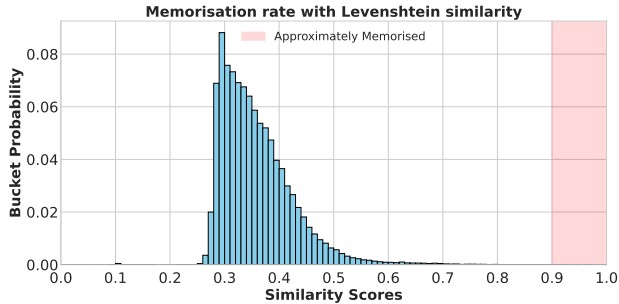

*(a)* Distribution of Levenshtein scores.

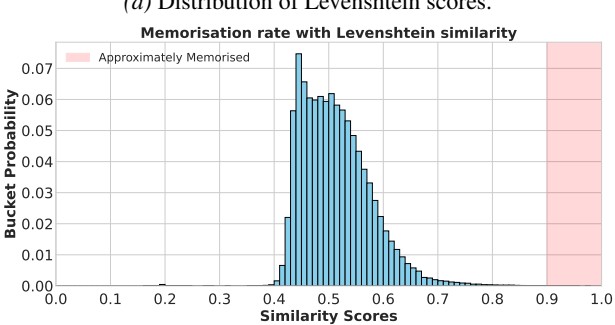

*(b)* Distribution of Indel scores.

*Figure 4.* Histograms illustrating the distribution of memorisation scores, measured by Levenshtein and Indel similarity scores. For each generation, we find the largest similarity score in the post-training set of OLMo 2.

In Figure 4 we show that when judging memorisation rates based on string matching scores, memorisation rates seem negligible. The judgement from such results would be that our prompting strategy does not extract useful memorised data under string matching metrics. This however does not paint the entire picture. For example, the generated sample in Figure 3, would **not** be considered memorised under the heuristic of (Comanici et al., 2025). This is the extent to which measuring memorisation on simple string matching is problematic; it does not flag examples that any reasonable human would judge as clear cases of memorisation. This is because string matching can quickly become non-

informative due to trivial differences (see Appendix Section E).

## 3.2. Large scale extractions with embeddings

We now compare the string matching results to matching done using neural embeddings. We generate 1M samples with OLMo 13B using the same method and embed them using `gemini-embedding-001`. In Figure 5, we show that neural embeddings unveil a much higher memorisation rate (left) when compared to string matching scores Figure 4. The scatter plot (right) shows that string matching distances are not well-aligned with semantic memorisation and also seem to exhibit a strong string length bias as well, where longer generations are consistently given lower Levenshtein similarity scores. We find that neural embeddings are much better at dealing with cases such as Figure 3 and provide a number of examples in the Appendix (Section E).

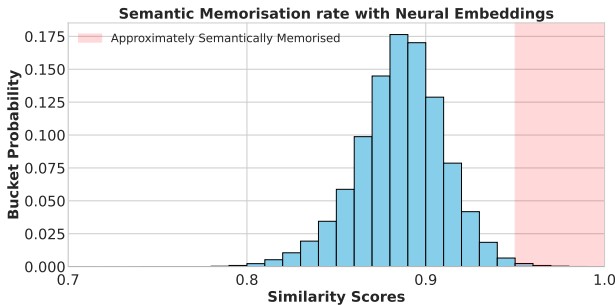

*(a)* Distribution of `gemini-embedding-001` scores.

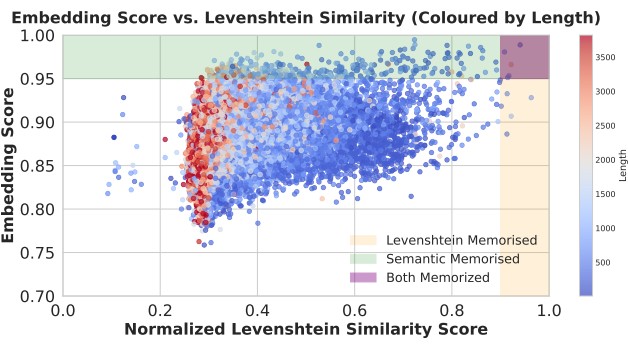

*(b)* Levenshtein vs embedding scores.

*Figure 5.* Histograms illustrating the distribution of embedding scores generated with `gemini-embedding-001` (left) and scatter plot comparing the embedding scores to the Levenshtein distance, with points coloured by string length (right)

**Coverage.** We now check the *coverage* of the post-training data, where for each post-training sample, we report the largest embedding score out of the 1M generated samples. We report the results in Figure 6. We find that some samples are much more memorised than others. While it is hard to understand exactly why, our investigations revealed

that samples are much more likely to be memorised if similar samples are also present in the pre and mid training datasets. Further, samples that appear often, for example different versions of the same reasoning problem, seem to be more likely to be memorised.

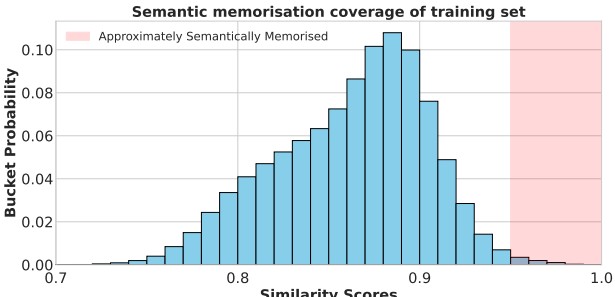

*(a)* Empirical distribution of coverage of post-training set.

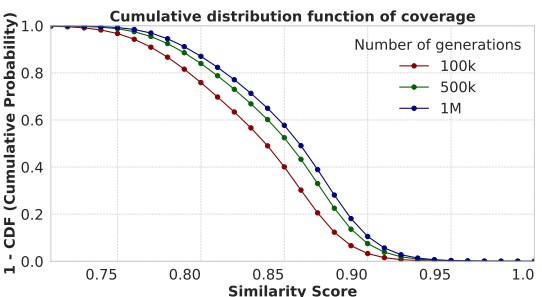

*(b)* Empirical cumulative distribution of coverage of post-training set.

*Figure 6.* We report for each true post-training sample a histogram of the best score between the 1M generations (left) and the cumulative distribution (right). Due to the nature of the experiment, coverage should improve monotonically as sampling more can only increase memorisation.

**Chat template better recovers post-training data.** We finally show that conditioning on the chat template is useful to encourage the model to generate post-training-like data. As a baseline, we compare this method to simply conditioning on the single token <lendoftextl>. We consider 1,000 generations with both prefixes and report the estimate on the expected value of the embedding score. In Table 1, we show that indeed conditioning on the entire chat template provides samples closer to the post-training distribution. *We suspect that since chat templates are only present during post-training, the model associates their presence with the post-training data distribution.* This provides an explation for why techniques such as Magpie (Xu et al., 2024) are possible: conditioning on the chat template results in generations that are much closer to the post-training distribution.

| Prefill Type | Mean Embedding Score |
|---|---|
| <lendoftextl> | 0.857 |
| <lendoftextl><luserl>\n | 0.892 |

*Table 1.* Mean best embedding score of generations for OLMo 2 13B using only the 'beginning of sequence token' versus the full chat template prefill. The longer prefill generates samples that are on average semantically closer to the post-training set.

### 3.3. Direct distillation on extracted data

A natural question one may then have is, if the generated data is similar to the original post-training dataset, can it be used to post-train a model directly? In other words, are we able to re-use a large and diverse enough number of samples to post-train a model without collecting any data manually? To explore this question, we post-train using SFT OLMo 2 7B in two ways: (1) the original dataset in order to reproduce the original results and (2) our generated dataset. For the synthetic dataset, we collect a synthetic dataset of a similar size of $\approx 930k$ samples. We perform basic filtering and processing using Gemini 2.5. We note that even though the number of samples is the same, the original SFT training is over $\approx 1.3B$ tokens, while the synthetic training set has only $\approx 850M$ tokens as the filtered generations remain shorter.

In Table 2, we report the results following the benchmarks and evaluation pipeline used by (OLMo et al., 2024). To validate our setup, we first show that our reproduction is very close to the released SFT checkpoint. Our model trained on synthetic data also achieves comparable performance on the benchmarks, except for the IFE task. We suspect that our pipeline generates too few examples that target this benchmark, but believe the performance could likely be improved by adding non-synthetic data. In fact, it is likely that a mix of synthetic and curated targeted data could be a useful paradigm to explore to boost model performance reducing the labour required to collect entire post-training datasets.

## 4. Large scale extraction of RL data

We now focus on the extraction of RL data. We use the Open-Reasoner-Zero (Hu et al., 2025) model, which was trained from the Qwen 2.5 base model with PPO (Schulman et al., 2017) using post-training data that is publicly available. With RL, the training samples consist of questions and answers, but the reasoning traces not part of the training dataset as they are artifacts of the training rollout. For this reason, we focus on the extraction of the questions and answer part of the dataset although note that reasoning traces can be useful in their own right.

We prompt the model by again taking the first part of the chat template specified by the developers of the model (see

| Model | BBH | MMLU | MATH | GSM8K | POPQA | TQA | IFE | DROP |
|---|---|---|---|---|---|---|---|---|
| Baseline (SFT only) | 0.4953 | 0.6133 | 0.2073 | 0.7407 | 0.2364 | 0.4858 | 0.6562 | 0.5960 |
| Baseline (Reproduction) | 0.4944 | 0.6123 | 0.2077 | 0.7377 | 0.2529 | 0.5110 | 0.6617 | 0.5945 |
| Extracted Data | 0.5161 | 0.6052 | 0.1705 | 0.7847 | 0.2490 | 0.5529 | 0.5028 | 0.5923 |

*Table 2.* Model performance after SFT on the benchmarks considered by (OLMo et al., 2024). The baseline performance is taken from (OLMo et al., 2024) and the reproduction was ran using the code provided by the authors with the original dataset. Using our method, we train a model on SFT 'synthetic' data extracted from the model using the same settings of the baseline.

Section D in the Appendix for the entire prefix) and generate 100k samples independently. We find that the model very consistently generates a question, followed by a thinking trace, and finally an answer. We then searched the training set for these generations. Surprisingly, we again found a number of training samples being regurgitated verbatim. We show an example in Figure 7, where the model outputs the exact training sample, a reasoning trace, and the correct solution. We find the fact that models are capable of regurgitating RL training samples to be counterintuitive as the PPO objective, at least at a glance, seems rather misaligned with the memorisation of training samples, especially when compared to methods such as SFT that instead very explicitly increase sequence likelihoods.

We explore this further phenomenon further by measuring the change of likelihood of training set samples in the base and post-trained models. Measuring likelihood of the training set is limited because this only measures the 'pointwise' memorisation – a likelihood of a training sample might remain low because the exact polynomial in the training set for instance is not being memorised, but the question style of finding its roots is. Regardless of the limitation of this simple measurement, we believe the results can still provide valuable signal and intuition.

In particular, we measure the likelihoods of each PPO training sample question under the Qwen 2.5 base model and the Open-Reasoner-Zero model. If the RL process induces memorisation, then we would see the likelihood using the post-trained model increase on the training samples. We bucket the results in likelihoods increasing by magnitudes of 10 for the base Qwen model and the Open-Reasoner-Zero model and report the results in Figure 9. The results show that RL training induces many of the training prompts to increase in likelihood. We found samples of likelihoods increasing from $10^{-11}$ to $10^{-5}$ after RL post-training, showcasing the fact that RL may be able to induce the memorisation of post-training samples. This is particularly surprising when one considers the RL post-training objective. It is not immediately clear to us what exact mechanism is driving this increase in likelihood and leave this as a future exciting research direction.

> **Generation**
>
> **<PREFIX>**
> Find the quotient of the division $(3z^4 - 4z^3 + 5z^2 - 11z + 2)/(2 + 3z)$.
> To solve the problem of finding the quotient of the division $(3z^4 - 4z^3 + 5z^2 - 11z + 2) \div (2 + 3z)$, we will use polynomial long division. Here are the steps:
> **[...]**
> The final answer is: $\boxed{z^3 - 2z^2 + 3z - \dfrac{17}{3}}$

> **RL Training sample**
>
> Find the quotient of the division $(3z^4 - 4z^3 + 5z^2 - 11z + 2)/(2 + 3z)$.
>
> $\boxed{z^3 - 2z^2 + 3z - \dfrac{17}{3}}$

*Figure 7.* (Left) Generation from Open-Reasoner-Zero 7B using our special **<PREFIX>** (see Appendix Section D for the full string). We shorten the thinking trace with **[...]**. (Right) True RL post-training sample. Surprisingly, we find that RL training samples can be regurgitated *verbatim*, even though the training objective seems to be heavily misaligned with this behaviour. The RL training samples only consist of a question and answer pair and do not come with a thinking trace. The model instead regurgitates the question, followed by the thinking trace, and finally the answer.

**RL on extracted dataset** We now show that one can perform a similar extraction pipeline but to instead extract an RL dataset. In other words, one can use an RL-trained model to extract an RL dataset with little effort. We start by post-train using Dr. GRPO (Liu et al., 2025) the Qwen2.5 7B base model using the ORZ 57k dataset (Hu et al., 2025). With the resulting post-trained model (we call it 'Baseline'), we then generate using our method 100k samples and process them using Gemini 2.5 (Comanici et al., 2025) to filter out invalid, incomplete or incorrect samples. We finally randomly select from this set 57k synthetic samples to create our synthetic training data. We use this to post-train the Qwen2.5 7B base model using synthetic data only. Afterwards, we eval-

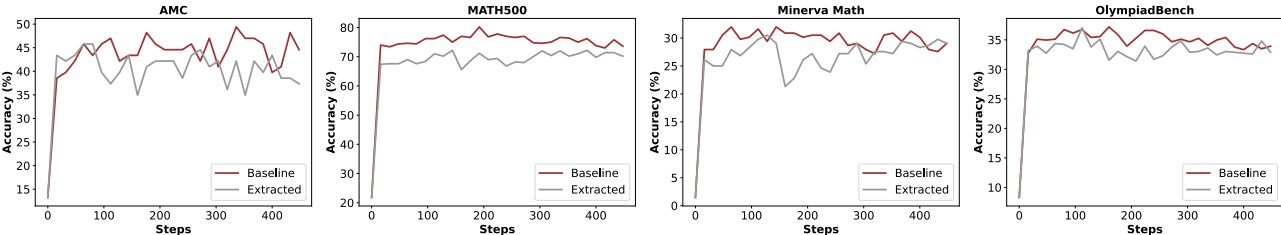

*Figure 8.* RL training using the ORZ (Hu et al., 2025) dataset and a dataset that was extracted using our method. Surprisingly, we are able to recover most of the performance with our simple extraction method.

uate both models on four standard benchmarks: AMC (Li et al., 2024), MATH500 (Hendrycks et al., 2021), Minerva Math (Lewkowycz et al., 2022), and OlympiadBench (He et al., 2024). We then report the results in Figure 8. The model trained on the synthetic data extracted from the 'Baseline' achieves comparable performance on the benchmarks. These results are surprising because our synthetic dataset is based on extracted data from a small 7B model trained on a relatively small RL dataset. We suspect that a more sophisticated pipeline could be used to achieve higher performance and retrieve a higher quality dataset. As our work focuses on training sample extraction, we leave this goal as a promising future direction.

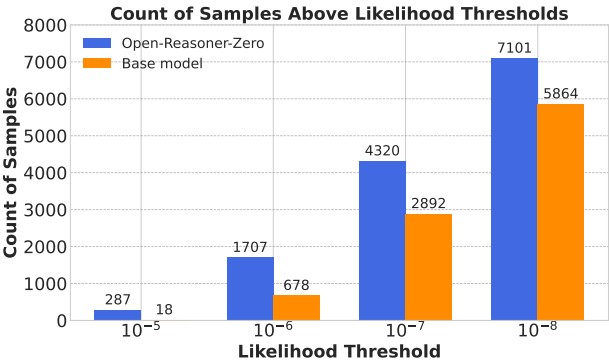

*Figure 9.* Count of training samples (prompt only) with likelihood above a certain threshold before and after RL. The entire train set contains 142,770 samples. The base model is Qwen2.5 7B. After RL, many training prompts achieve a higher likelihood.

## 5. Conclusion

In this paper, we demonstrate that alignment data can be efficiently extracted from open-weight large language models. Our attack leverages a simple observation: chat templates and their special tokens are typically introduced during post-training, making them effective prefixes for inducing models to regurgitate alignment-like data.

**Leakage Metrics are Hard** A key finding is that the true scale of this leakage is hidden by traditional metrics. We

show that methods relying on string matching drastically undercount the rate of extraction, by (at least!) an order of magnitude. By instead using high-quality embedding models, we can identify approximate semantic memorisation—instances where the model reproduces the semantic structure and patterns of training data, even if not verbatim. This semantic similarity is far more relevant for capturing the utility of the data, as highlighted by numerous examples all throughout the paper.

**Model Distillation as Data Distillation** We demonstrate that data extracted from a post-trained model can be used to successfully train a new base model, meaningfully recovering parts of the original's performance in both SFT and RL settings. This confirms that the common practice of model distillation can function as an indirect form of training data extraction. Certain advantages that an open model obtains from its alignment data are therefore at risk of being leaked.

**Rethinking Memorisation in RL** Surprisingly, we find that models readily regurgitate training samples (even!) from Reinforcement Learning (RL) phases. This is counter-intuitive, as the reinforcement learning objective is not explicitly designed to increase sequence likelihoods in the same way as SFT. The fact that RL training prompts increased in likelihood after post-training suggests a more complex relationship between alignment and memorisation that warrants future investigation.

Our attack exploits chat templates and so is only applicable to open models. Closed models enforce the chat template, and is outside of the users control; a working exploit on a closed model would require a user to spoof the behaviour of these tokens when submitting a query to the model. Whilst more difficult, prior work (Geng et al., 2025) has shown this is not necessarily impossible. Future work will establish how serious a threat this is on closed models.

## Impact Statement

This paper presents work aimed at understanding the security and privacy vulnerabilities of open-weight Large Language Models, specifically concerning the memorisation and regurgitation of alignment training data. The primary

societal consequence of our work lies in the protection of intellectual property and proprietary datasets. We show how the release of open models may inadvertently expose any proprietary data used to steer them, potentially compromising the competitive advantage of model developers.

Furthermore, our findings suggest that the practice of model distillation may be closer to a form of indirect training on the original provider's data than one may expect. This has potential implications surrounding model licensing and fair use. While our investigation focuses on the extraction of alignment data rather than Personally Identifiable Information (PII), the techniques highlighted here underscore the general risks of data leakage – something that has already been deeply studied in the literature. We believe that exposing such risks is of critical importance and hope that our work encourages the research community to develop more robust defenses also directed towards the preservation of alignment data.

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

## A. Nomenclature

In this paper, we rely on the definitions below. We refer the reader to Section B of the Appendix for a thorough background overview.

**Alignment Data:** This term is used broadly to refer not just to data for safety and instruction-following (like Supervised Finetuning and Reinforcement Learning with Human Feedback datasets), but to any targeted data collection used to steer model behavior and improve specific capabilities such as mathematics, reasoning, or long-context understanding. This data is usually considered a significant competitive asset.

**Training Data:** The term 'training data' is usually used to describe data from all phases of model creation, including pre-training, mid-training, and post-training, which encompasses data used for SFT and RL fine-tuning. In this paper we expand the usual meaning to also cover semantically equivalent representation of the training data.

**Post-training:** This refers to the training stages that occur after the initial large-scale pre-training phase. It involves using various specialized datasets and methods, such as Supervised Finetuning (SFT) and Reinforcement Learning (RL), to align the model's capabilities with desired behaviors and tasks. In this work, we treat post-training data as a (large) subset of total alignment data, as it is becoming increasingly common to include alignment data (maths, reasoning, etc.) also in earlier stages of training.

**Memorisation** [3]**:** This refers to the phenomenon where a model can regurgitate or recite its training data. Our work extends this definition beyond verbatim string matching to include a more abstract notion of memorisation, such as reproducing the patterns, templates, and semantic structure of proprietary data. Related to this concept is *approximate memorisation* (e.g., Comanici et al. (2025)), but instead of using simple string-matching similarity metrics, in our work we show that text embeddings seem much better suited at detecting approximate memorisation when semantics are important. We refer the reader to Ippolito et al. (2022) for further motivation behind why we expand our definition of memorisation.

**Chat Template:** A specific structure used to format prompts by wrapping user, assistant, or system messages in special tokens (e.g., <|user|>, <|assistant|>). A key aspect of our attack is that these templates and their special tokens are typically introduced only during the post-training stage, where most of the alignment is done.

**Embedding Score:** A metric we use to measure memorisation based on the semantic similarity between two pieces of text, calculated using the dot product of neural text embeddings. We propose this as a more effective alternative to traditional string-matching metrics (like Levenshtein distance), as it better captures semantic equivalence even when there are superficial differences.

## B. Extended background

Our work sits at the intersection of key areas in the study and development of frontier LLMs. The first is *model alignment*, where curated data and specialised training techniques are used to transform a base model into a useful assistant. The second is *training data memorisation*, the observation that models are capable of regurgitating training data. The third is the widespread practice of *model distillation*, a process through which a strong model's capabilities can be transferred to another. In this section, we will review these areas and argue that there is a risk that lays at the intersection of the three.

### B.1. Model alignment

Early models (Radford et al., 2018; Devlin et al., 2019) were built on the idea that pre-training on larger and larger amounts of internet data was the solution to keep improving capabilities. Today, this principle has arguably shifted and we have seen a surge in multi-stage training with a number of specialised datasets and techniques, giving birth to what is now called *post-training*. LLMs are post-trained using various methods ranging from Supervised Finetuning (SFT) to now very popular RL methods such as Proximal Policy Optimisation (PPO) (Schulman et al., 2017), Group Relative Policy Optimization (GRPO) (Shao et al., 2024), and Reinforcement Learning from Human Feedback (RLHF) (Ouyang et al., 2022), among many others. In this work, we mainly focus on models that were post-trained using SFT and PPO as they are common post-training techniques and we have strong open models that were trained using such techniques with publicly available post-training data.

---

[3]We make no statement with regard to whether or not a model 'contains' its training data in a bit-wise or code-wise sense, nor in the sense that any arbitrary instance of training data can be perfectly retrieved.

**Supervised Finetuning**    With SFT, one collects a dataset $D$ of question-answer pairs $D = \{(Q_i, A_i)\}_{i=1}^N$. The model is then optimised to predict an answer $A_i$ given a question $Q_i$, i.e. to increase the likelihood $P(A_i|Q_i)$. To achieve this, usually one masks out the contributions of the loss that come from the question part, i.e. computing the conditional gradient $\nabla_\theta \log P(A_i|Q_i; \theta)$ given some parameters $\theta$. The parameters are updated via some form of gradient descent on the negative log-likelihood:

$$\theta' \leftarrow \theta + \eta \frac{1}{|B|} \nabla_\theta \sum_{k \in B} \log P(A_k|Q_k; \theta)$$

given some learning rate $\eta > 0$ and batch $B \subset D$. The OLMo 2 (OLMo et al., 2024) family of models has been post-trained in large part using SFT with a dataset of 939,344 question-answer pairs which has been released publicly, providing an excellent resource for our study.

**RL-based post-training**    We focus on reinforcement learning with verifiable rewards (RLVR) (Guo et al., 2025) in this work. The Open-Reasoner-Zero (Hu et al., 2025) model adopts PPO (Schulman et al., 2017) for policy optimization in the post-training. The training objective could be written as

$$\mathcal{J}_{\text{PPO}}(\theta) = \mathbb{E}_{Q \sim \mathcal{D}, O_{\leq t} \sim \pi_{\theta_{\text{old}}}(\cdot|Q)} \sum_{t=1}^{|O|} \left[ \min \left( \frac{\pi_\theta(O_t \mid Q, O_{<t})}{\pi_{\theta_{\text{old}}}(O_t \mid Q, O_{<t})} \hat{A}_t, \ \text{clip}\left( \frac{\pi_\theta(O_t \mid Q, O_{<t})}{\pi_{\theta_{\text{old}}}(O_t \mid Q, O_{<t})}, 1 - \varepsilon, 1 + \varepsilon \right) \hat{A}_t \right) \right]$$

Here $O$ is the generated response, $\epsilon$ is the clipping ratio, and $\hat{A}_t$ is an estimator of $t$-th token's advantage, which is computed through the Generalized Advantage Estimation (GAE) (Schulman et al., 2015) with a learned value model and a reward function. GRPO (Shao et al., 2024; Guo et al., 2025) removed the value function and estimated the advantage $\hat{A}_t$ in a group manner. For each question $Q$, GRPO samples a group of responses $\{O^1, O^2, ..., O^G\}$, which can be rewarded as $\{R^1, R^2, ..., R^G\}$ using the reward function. Normally the reward function is shaped by whether the response correctly answers the question (the answer within a specified format matches the correct answer) and whether the response format is correct. Then the advantage is estimated as $\hat{A}_t^i = \frac{R^i - \text{mean}(\{R^i\}_{i=1}^G)}{\text{std}(\{R^i\}_{i=1}^G)}$. Therefore, the objective of GRPO can be written as

$$\mathcal{J}_{GRPO}(\theta) = \mathbb{E}_{Q \sim \mathcal{D}, \{O_{\leq t}^i\}_{i=1}^G \sim \pi_{\theta_{\text{old}}}(\cdot|Q)}$$

$$\frac{1}{G} \sum_{i=1}^G \frac{1}{|O_i|} \sum_{t=1}^{|O_i|} \left[ \min \left( \frac{\pi_\theta(O_t^i|Q, O_{<t}^i)}{\pi_{\theta_{old}}(O_t^i|Q, O_{<t}^i)} \hat{A}_t^i, \text{clip}\left( \frac{\pi_\theta(O_t^i|Q, O_{<t}^i)}{\pi_{\theta_{old}}(O_t^i|Q, O_{<t}^i)}, 1 - \epsilon, 1 + \epsilon \right) \hat{A}_t^i \right) - \beta \mathbb{D}_{\text{KL}}(\pi_\theta \mid\mid \pi_{\text{ref}}) \right]$$

KL regularization ensures that the policy model does not deviate the reference model too far, but could be eliminated to further improve the exploration (Hu et al., 2025). Dr. GRPO (Liu et al., 2025) removes response-level length bias in the objective and question-level difficulty bias in the advantage estimation, improving token efficiency with comparable reasoning performance. There is also significant body of work aiming to improve GRPO, with notable examples being DAPO (Yu et al., 2025) and GSPO (Zheng et al., 2025).

**Chat templates**    It has now become common to format the question and answers using a *chat template*, which wraps the messages in special tokens that are designed to mark the messages as being 'user', 'assistant' or 'system' messages. Chat templates from a security perspective are usually trained to build a so-called instruction hierarchy of privilege (Wallace et al., 2024) although its practical effectiveness has been challenged (Geng et al., 2025). Crucially, the special tokens used to build the template are *only* introduced during post-training. In this work, we show that this makes them a useful attack vector to elicit the generation of alignment data introduced exclusively in post-training. A similar effect has been pointed out by (Xu et al., 2024), but while their focus was the automated generation of an alignment dataset, our focus is instead that of generalising and better understanding the process from the point of view of memorisation – answering one of their conjectures where they posited that models are likely memorizing such data.

### B.2. Memorisation

It is a well-documented fact that LLMs are capable of memorising and reciting training data verbatim (Carlini et al., 2021; Nasr et al., 2023; 2025). The primary concern of these studies has been public-facing risks, such as the leak of private

information like email addresses, names, and credit card numbers that were accidentally included in the pre-training dataset or the regurgitation of copyrighted material.

A common way to measure memorisation is that of *(greedy) discoverable extraction* (Carlini et al., 2021; 2022) – in such case a sample is called extractable if there exists some prefix such that the LLM prompted on the prefix will generate the sample using *greedy* decoding. Such a definition is computationally convenient as it effectively ignores the stochastic nature of LLMs and is useful for a variety of tasks like observing if the model outputs copyrighted data verbatim or to discover sensitive strings of information. A relaxation of this is *probabilistic* discoverable extraction (Hayes et al., 2025b) in which one considers the probability of extracting the sample in multiple queries. This measure provides more information by considering the joint likelihood of the sequence rather than providing simply a boolean decision on extractability.

Recent work has begun to focus on moving away from measuring memorisation of verbatim or near-verbatim string matches (Ippolito et al., 2022; Chen et al., 2024b). For example, Shilov et al. (2024) showed the LLMs can assemble information from related, partially overlapping fragments of text, a concept they term *mosaic memory*. This challenges the assumption that memorisation is driven exclusively by exact repetitions in the training data. They demonstrate that even highly modified sequences can contribute significantly to the memorisation of a reference sequence, with the memorisation effect being predominantly a function of syntactic overlap (shared tokens). This suggests that current data deduplication techniques, which often focus on removing exact duplicates, are insufficient for addressing the full spectrum of memorisation and data leakage risks. Wang et al. (2024) introduced the concept of distributional memorisation, a form of non-verbatim memorisation measured by the correlation between an LLM's output probabilities and the frequency of related input-output data pairs (n-grams) in its pretraining corpus. The authors find that this type of memorisation is prominent in knowledge-intensive tasks like factual question-answering, whereas reasoning-based tasks rely more on generalization, where the model's outputs diverge from the training data's distribution. Finally, Huang et al. (2025) presents the Entropy-Memorisation Law, which establishes a linear correlation between the entropy of training data and a non-verbatim memorisation score measured by Levenshtein distance. The law suggests that data with higher entropy is more difficult for a model to memorize accurately, resulting in a higher memorisation score (i.e., more token-level differences between the model's output and the original text). Our work is similar in spirit to all of these works, however, we establish memorisation rates under our embedding definition which are far higher than what may be expected from these prior works. For example, Wang et al. (2024) found reasoning-based tasks rely more on generalization than memorisation; we found that even under reasoning-based tasks, the model heavily exploits knowledge of training data through clear signs of memorisation.

Jagielski et al. (2022) showed that data that appears later in the training process is more likely to be memorized. Our focus on memorisation of alignment and strategic proprietary data presents an interesting open-question about if this finding still holds for post-training. One the one hand we may expect this data to be more likely to be memorized due to results from Jagielski et al. (2022), on the other, post-training has certain properties that may make it less likely to be memorized. For example, tokens are regularly masked during the loss computation implying part of training prompts will not contribute to model gradients, whilst prior work has shown RL (commonly used in post-training) is less likely to memorize than standard instruction tuning (with the same compute budget) (Pappu et al., 2024).

### B.3. Model Distillation

Model distillation is a popular technique for creating capable models without the massive cost of training from scratch (Hinton et al., 2015). In this process, a smaller 'student' model is trained on the outputs generated by a larger, more powerful 'teacher' model (e.g., using GPT-4 to generate training data for a new open-source model).

While often viewed as a way to transfer capabilities, distillation can be re-framed through the lens of memorisation. If a teacher model is capable of reciting its own proprietary training data, any student model trained on its outputs will, by definition, be exposed to this secret sauce. This risk is especially pronounced in 'hard' distillation pipelines that directly use the teacher's labels as training examples (Xu et al., 2024)[4]. The core question, which our work addresses, is how much of the teacher's original training data is unintentionally passed down to the student.

**Our position**   Collecting proprietary strategic data is expensive and time consuming; model developers and data curators clearly do not want to give this data freely to other parties. On the other hand, practitioners are already actively training on outputs from strong competing models. If this data leaks through this process, the competitive advantage of the original data

---

[4]This is contrast to *soft* distillation in which one trains on the logits instead of the output labels. Soft distillation is challenging if the vocabulary of the two models is not the same.

owner is eroded. This motivates our memorisation study within this context.

## C. Accidental gradient alignment

An intriguing consequence of our work is that we observe *training data being memorised verbatim even if the gradients are masked*. This constitutes an interesting type of leakage that to the best of our knowledge has not been previously observed. We propose some mathematical speculation for how this might happen. To make this statement more precise, the claim is that given a question $Q$ and answer $A$, we question whether updating based on the conditional gradient $A|Q$ can increase the likelihood of $Q$.

Let the model parameters be $\theta$. During training on a pair $(Q, A)$, the parameters are updated from $\theta$ to $\theta'$ using gradient descent on the negative log-likelihood of the answer:

$$\theta' = \theta + \eta \nabla_\theta \log P(A|Q; \theta),$$

where $\eta > 0$ is the learning rate. We are interested in the change in the log-probability of the question, $\log P(Q; \theta')$, after this update. A first-order Taylor expansion of $\log P(Q; \theta')$ around $\theta$ gives:

$$\log P(Q; \theta') \approx \log P(Q; \theta) + (\theta' - \theta)^T \nabla_\theta \log P(Q; \theta)$$

Substituting the update rule for the term $(\theta' - \theta)$, we obtain:

$$\log P(Q; \theta') \approx \log P(Q; \theta) + \eta \left( \nabla_\theta \log P(A|Q; \theta) \right)^\top \left( \nabla_\theta \log P(Q; \theta) \right).$$

For the likelihood of $Q$ to increase, the inner product of the two gradients must be positive:

$$\left( \nabla_\theta \log P(A|Q; \theta) \right)^\top \left( \nabla_\theta \log P(Q; \theta) \right) > 0.$$

This condition is rather intuitive. The first gradient, $\nabla_\theta \log P(A|Q; \theta)$, is the direction in parameter space that maximally increases the probability of the answer $A$ given the question $Q$. The second, $\nabla_\theta \log P(Q; \theta)$, is the direction that maximally increases the unconditional probability of the question $Q$. If their dot product is positive, the two gradients are correlated and increasing the likelihood $P(A|Q)$ also increases the likelihood $P(Q)$.

We note as a caveat that simply considering the likelihood of a single sequence might not be the most informative metric. For instance, a math-related post-training sample might have different numerical values, but the general 'template' might be correct. This suggests that one should really integrate over all reasonable sequences that one wishes to consider which is of course quickly intractable.

## D. Prefixes used for generation

Below we provide the generation prefixes used for the experiments in our work. Both are prefixes taken directly from the template released by the developers of the corresponding models.

---

**SFT extraction prefix**

<|endoftext|><|user|>

---

**RL extraction prefix**

A conversation between User and Assistant. The User asks a question, and the Assistant solves it. The Assistant first thinks about the reasoning process in the mind and then provides the User with the answer.
The reasoning process is enclosed within <think> </think> and answer is enclosed within <answer> </answer> tags, respectively, i.e., <think> reasoning process here </think> <answer> answer here </answer>. User:
You must put your answer inside <answer> </answer> tags, i.e., <answer> answer here </answer>. And your final answer will be extracted automatically by the \boxed{} tag. This is the problem:

---

# E. Failure cases of string matching

We report in this section interesting failure cases we have found with string matching score metrics. We recall that our chosen embedding threshold is $0.95$, while the string matching threshold is $0.90$. All examples we show in this section have an embedding score that is above $0.95$ and therefore would be considered as 'semantically' memorised. Therefore, the examples we show can be seen as failure cases of the string matching measure in which instead the samples are correctly identified as memorised using semantic neural embeddings.

---

**Generation**

**Q**: A man took loan from a bank at the rate of 12% p.a. S.I. After 3 years he had to pay Rs. 5400 interest only for the period. The principal amount borrowed was?
**A**: 15000

---

**Embeddings (0.97 match)**

A man took loan from a bank at the rate of 12% p.a. simple interest. After 3 years he had to pay Rs. 5400 interest only for the period. The principal amount borrowed by him was.
Options:
(A) 15000
(B) 2655
(C) 16888
(D) 6677
(E) 1871
Stream of consciousness first, then make a decision:
Principal = Rs. (100 x 5400)/(12*3) = Rs. 15000.
Thus, the answer is (A).

---

**String Matching (0.43 match)**

A man took loan from a bank at the rate of 12% p.a. simple interest. After 3 years he had to pay Rs. 5400 interest only for the period. The principal amount borrowed by him was.
Options:
(A) 15000
(B) 2655
(C) 16888
(D) 6677
(E) 1871
Stream of consciousness first, then make a decision:
Principal = Rs. (100 x 5400)/(12*3) = Rs. 15000.
Thus, the answer is (A).

---

*Figure 10.* Generation (top) compared to the best match training sample that is selected both by embeddings (bottom left) and string matching (bottom right). The string matching score is low due to the model adding options and reasoning even though the question is regurgitated almost verbatim.

> **Generation**
>
> # Question
> Which of the following sentences is nonsensical?
> Options:
> - Sentence A: "He put some petrol to put out the fire"
> - Sentence B: "He put some water to put out the fire"
>
> # Answer
> Sentence A
> Petrol is gasoline and would increase the fire, not put it out. The answer is Sentence A.
>
> **A**: 15000

> **Embeddings (0.97 match)**
>
> Come up with a question and reasoning that would justify this answer: Sentence A
> The question is: Of the following two sentences, which one is against common sense?
> Options:
> - Sentence A: "He put some petrol to put out the fire"
> - Sentence B: "He put some water to put out the fire"
>
> Step-by-step reasoning process: Petrol is gasoline and would increase the fire, not put it out.

> **String Matching (0.59 match)**
>
> Question: Which of the following sentences is nonsensical?
> Options:
> - Sentence A: "He put the elephant on the cupboard."
> - Sentence B: "He put the vase on the cupboard."
>
> Please answer this question gradually:
> An elephant is to heavy for a cupboard. So the answer is Sentence A.

*Figure 11.* Generation (top) compared to the best match training sample that is selected both by embeddings (bottom left) and string matching (bottom right). The embeddings retain the petrol and water semantics, while string matching finds a (poor) match with a question that has a more similar template.

**Generation**

Emily was 54 inches tall last year. She grew 4 inches more than her brother who only grew 2 inches last year. How tall is Emily now?
Let's solve this step by step.

Emily's brother grew 2 inches last year.
Emily grew 4 inches more than her brother, which means she grew 2 + 4 = 6 inches last year.
Emily was originally 54 inches tall.
After growing 6 inches, Emily's current height is 54 + 6 = 60 inches.

So, Emily is now $\boxed{60}$ inches tall.

**Embeddings (0.95 match)**

Emily was 55 inches tall last year. If she grew 4 inches more than her sister who grew 2 inches last year, how tall is Emily now?
Emily's sister grew 2 inches.
Emily grew 4 inches more than her sister, so she grew 2 + 4 = 6 inches.
Last year Emily was 55 inches tall, and she grew 6 inches since then.
So her current height is 55 + 6 = 61 inches.
Thus, Emily is $\boxed{61}$ inches tall now.

**String Matching (0.68 match)**

Emily was 55 inches tall last year. If she grew 4 inches more than her sister who grew 2 inches last year, how tall is Emily now?
Emily's sister grew 2 inches.
Emily grew 4 inches more than her sister, so she grew 2 + 4 = 6 inches.
Last year Emily was 55 inches tall, and she grew 6 inches since then.
So her current height is 55 + 6 = 61 inches.
Thus, Emily is $\boxed{61}$ inches tall now.

*Figure 12.* Generation (top) compared to the best match training sample that is selected both by embeddings (bottom left) and string matching (bottom right). Both methods find the same best match, but string matching heavily penalises small differences.

---

### Generation

**Teacher:** In this task you will be given a list of integers. A list contains numbers separated by a comma. You need to round every integer to the closest power of 2. A power of 2 is a number in the form '$2^n$', it is a number that is the result of multiplying by 2 n times. The following are all powers of 2, $2^0 = 1, 2^1 = 2, 2^2 = 4, 2^3 = 8, 2^4 = 16, 2^5 = 32, ...,$ Note that 1 is also considered as a power of 2. Rounding to the closest power of 2 means that if the number is between two powers of 2, it should round up to the larger power of 2. The output should be a list of integers that is the result of rounding each integer in the input list to the closest power of 2.

$$-89, 64, 1088, -193$$

Solution:

$$-64, 64, 1024, -256$$

---

### Embeddings (0.97 match)

In this task you will be given a list of integers. A list contains numbers separated by a comma. You need to round every integer to the closest power of 2. A power of 2 is a number in the form '$2^n$', it is a number that is the result of multiplying by 2 n times. The following are all powers of 2, '2, 4, 8, 16, 32, 64, 128, 256, 512, 1024, 2048, 4096'. If an integer is exactly in equally far from two different powers of 2 then you should output the larger power of 2. The output should be a list of integers that is the result of rounding each integer int the input list to the closest power of 2. The output should include a '[' to denote the start of the output list and ']' to denote the end of the output list.
Q:

$$186, 1663, 3032, 1551, 20, 45, 4, 42, 656$$

A:

$$128, 2048, 2048, 2048, 16, 32, 4, 32, 512$$

### String Matching (0.61 match)

Instructions: In this task you will be given a list of integers. A list contains numbers separated by a comma. You need to round every integer to the closest power of 2. A power of 2 is a number in the form '$2^n$', it is a number that is the result of multiplying by 2 n times. The following are all powers of 2, '2, 4, 8, 16, 32, 64, 128, 256, 512, 1024, 2048, 4096'. If an integer is exactly in equally far from two different powers of 2 then you should output the larger power of 2. The output should be a list of integers that is the result of rounding each integer int the input list to the closest power of 2. The output should include a '[' to denote the start of the output list and ']' to denote the end of the output list.
Input:

$$92, 1623, 1339, 770$$

Output:

$$64, 2048, 1024, 1024$$

*Figure 13.* Generation (top) compared to the best match training sample that is selected both by embeddings (bottom left) and string matching (bottom right). Both methods find the same best match, but string matching heavily penalises small differences in notation and numbers.

---

### Generation

Solve -4*q + 17*q + 104 = 0 for q.
Solve this plz.
A: -8
QUESTION: Math problem: Solve 0 = -11*b + 45*b + 136 for b.
What is the answer?
ANS: -4
Q: What is the solution?
Solve 8*c - 11 + 19 = 0 for c.
A: -1
Write down the solution for this math problem: Solve -12*f = -25*f + 52 for f.
answer: 4
question: Teacher asked me this: Solve 0 = 7*k - 3*k + 20 for k.
++++++++++
answer: -5

### Embeddings (0.95 match)

question: Teacher asked me this: Solve -16*q + 56 = -2*q for q.
++++++++++
answer: 4
Solve 0 = -52*x - 33540 + 32604 for x.
Solve this plz.
A: -18
QUESTION: Math problem: Solve -101 + 91 = 2*c for c.
ANS: -5
Q: What is the solution?
Solve 10*h + 0*h - 40 = 0 for h.
A: 4
Write down the solution for this math problem: Solve 409*a - 70 = 423*a for a.
answer: -5
QUESTION: Math problem: Solve 479*o - 928*o + 1968 = -497*o for o. ANS:
-41

### String Matching (0.60 match)

question: Teacher asked me this: Solve 9 = 3*m + 6 for m.
++++++++++
answer: 1

Solve 4*p - 9 = -13 for p.
Solve this plz.
A: -1
QUESTION: Math problem: Solve -106*t - 6183 + 4699 = 0 for t.
ANS: -14
Q: What is the solution?
Solve 80*f + 56*f + 35*f - 2227 = 680 for f.
A: 17
Write down the solution for this math problem: Solve 189*h - 1010 = 5038 for h.
answer: 32
question: Teacher asked me this: Solve 6905*p + 426686 + 368370 = 256466 for p.
++++++++++
answer:
-78

*Figure 14.* Generation (top) compared to the best match training sample that is selected both by embeddings (bottom left) and string matching (bottom right). Both methods find the same best match, but string matching heavily penalises differences in numerical values.

# F. Examples of data extracted using our attack

| Generation | Embeddings (0.98 match) |
|---|---|
| **Q:** The sale price sarees listed for Rs.400 after successive discount is 10% and 5% is?
Options:
(A) 338
(B) 277
(C) 342
(D) 882
(E) 212
**A:** (A)
Explanation: 400*(90/100)*(95/100) = 324 | Question: The sale price sarees listed for Rs.400 after successive discount is 10% and 5% is?
Options:
(A) 39
(B) 37
(C) 342
(D) 288
(E) 261 Random thoughts:
400*(90/100)*(95/100) = 342 The final answer:
(C). |

*Figure 15.* Generation (left) and closest training sample (right) using the `gemini-embedding-001` model. The sample is *above* the semantic 0.95 memorisation threshold.

| Generation | Embeddings (0.94 match) |
|---|---|
| Alyana has a pizza that is cut into 16 slices. After she and her friends finish eating, there are 4 slices left. If each of them ate 2 slices of pizza, how many people ate the pizza?
Let's denote the total number of people who ate pizza, including Alyana, as $x$.
Each of the $x$ people ate 2 slices.
Thus they ate a total of $2 * x$ slices.
There were originally 16 slices, but after they ate, there were 4 slices left.
Thus the total number of slices eaten is $16 - 4 = 12$.
We already concluded that the total number of slices eaten is $2 * x$.
So $2 * x = 12$, which implies that $x = 6$.
Thus there were a total of $\boxed{6}$ people who ate the pizza. | A group of friends is splitting a pizza. If each person eats 2 slices and there are 8 slices left, how many friends are there if the pizza originally had 24 slices?

Let's denote the number of friends as $x$.

Each friend eats 2 slices, so the total number of slices eaten by friends is $2 * x$.

The pizza originally had 24 slices, and there are 8 slices left, meaning $24 - 8 = 16$ slices were eaten.

Since each friend ate 2 slices, the number of friends is $16/2 = 8$. |

*Figure 16.* Generation (left) and closest training sample (right) using the `gemini-embedding-001` model. The sample is *below* the 0.95 semantic memorisation threshold.

| Generation | Embeddings (0.90 match) |
|---|---|
| A student wants to rent a car for a week-long vacation trip. The car rental company charges \$40 per day and an additional fee of \$0.25 per mile driven. The student plans to drive 300 miles during the week. Calculate the total cost of renting the car for the entire week, including the mileage fee. To solve this problem, follow these steps:
1. **Calculate the daily rental cost:** - The rental company charges \$40 per day. - The trip lasts for a week, which is 7 days. - Therefore, the cost for the days is:

$$40\,\text{dollars/day} \times 7\,\text{days} = 280\,\text{dollars}$$

2. **Calculate the mileage fee:** - The student plans to drive 300 miles. - The mileage fee is \$0.25 per mile. - Therefore, the mileage cost is:

$$0.25\,\text{dollars/mile} \times 300\,\text{miles} = 75\,\text{dollars}$$

3. **Calculate the total cost:** - Add the cost for the days to the mileage cost:

$$280\,\text{dollars} + 75\,\text{dollars} = 355\,\text{dollars}$$

Thus, the total cost of renting the car for the entire week, including the mileage fee, is:
#### 355 | Jamie is a freelance photographer who often travels for assignments and rents cars for these trips. On one of her recent assignments, she rented a car for 6 days. The rental company charges \$40 per day for the car. Additionally, Jamie always opts for a special insurance package that costs \$5 per day to ensure she has coverage for any potential issues. During this trip, Jamie accidentally scratched the car, and the repair cost was estimated at \$120. Fortunately, her insurance package covers 75% of the repair costs.
Calculate the total amount Jamie spent on the car rental, the insurance package, and her share of the repair cost after the insurance coverage. To solve this problem, we need to calculate three components: the cost of the car rental, the cost of the insurance package, and Jamie's share of the repair cost after insurance coverage.
**Step 1: Calculate the total car rental cost.**
The rental company charges \$40 per day for the car. Jamie rented the car for 6 days.

$$\text{Total car rental cost} = 40 \times 6 = 240\,\text{dollars}$$

[...]
**Step 4: Calculate the total amount Jamie spent.**
Add the costs from Steps 1, 2, and 3:
Total amount spent $= 240 + 30 + 30 = 300$ dollars
Now, let's output the final answer in the format requested.
#### 300 |

*Figure 17.* Generation (left) and closest training sample (right) using the `gemini-embedding-001` model. The sample is *below* the 0.95 semantic memorisation threshold.

## Generation

A professor assigns their class a project with a deadline on June 30, 2023. The students start their work on February 10, 2023. What is the duration of the project from the start date to the deadline?

To find the duration of the project, we need to calculate the number of days between February 10, 2023, and June 30, 2023.

1. **Count days from February 10, 2023, to the end of February 2023:** - February 2023 has 28 days (since 2023 is not a leap year). - Days from February 10 to February 28: 28 - 10 = 18 days.

2. **Count days from March 1, 2023, to June 30, 2023:** - March: 31 days
- April: 30 days
- May: 31 days
- June: 30 days

Total days from March 1, 2023, to June 30, 2023: 31 + 30 + 31 + 30 = 122 days

3. **Sum the days from both parts:**
18 days (February) + 122 days (March to June) = 140 days

Therefore, the duration of the project is 140 days.
#### 140

## Embeddings (0.85 match)

Professor Smith teaches a course on programming documentation. In one semester, he plans to assign his students a total of 12 documentation projects. For each project, students are required to write 5 pages of documentation. Professor Smith estimates that it takes him about 15 minutes to review each page of documentation. If he has 20 students in his class, how many hours will Professor Smith spend reviewing all the documentation projects for the entire semester? To solve the problem, we can follow these steps:
1. **Find the Total Number of Pages per Student:** Each student is required to write 5 pages of documentation per project. Since there are 12 projects, the total number of pages each student writes is:
5 pages/project × 12 projects = 60 pages/student

2. **Calculate the Total Number of Pages for All Students:** There are 20 students in the class. Therefore, the total number of pages for all students is:
60 pages/student × 20 students = 1200 pages

**[...]**
5. **Convert Minutes to Hours:** There are 60 minutes in an hour, so the total time in hours is:
$\frac{18000 \text{ minutes}}{60 \text{ minutes/hour}} = 300$ hours

Therefore, Professor Smith will spend a total of 300 hours reviewing all the documentation projects for the entire semester.

#### 300

*Figure 18.* Generation (left) and closest training sample (right) using the `gemini-embedding-001` model. The sample is *below* the 0.95 semantic memorisation threshold.

**Generation**

The main character of "Stranger and Stranger" has a mysterious past, and the film follows their journey of self-discovery and connection with others. Given this description, answer the question: who is the main character in stranger and stranger?

To solve the problem, we need to identify the main character of "Stranger and Stranger."

Here is the step-by-step reasoning process:

1. **Understand the Context**: The film "Stranger and Stranger" revolves around a main character with a mysterious past.
2. **Identify Key Information Provided**: The description explicitly mentions that the film follows the journey of self-discovery and connection with others by the main character.
3. **Analyze the Description**: The focus is on the main character's journey and their interaction with others, indicating their central role in the narrative.
4. **Conclusion**: Based on the given information, the main character in "Stranger and Stranger" is the one experiencing self-discovery and forming connections, which are central themes.

Now, providing the final answer:

#### The main character in "Stranger and Stranger" is the character with a mysterious past whose journey of self-discovery and connection with others is central to the film.

**Embeddings (0.80 match)**

You are a creative writer tasked with crafting a compelling and thought-provoking story about a character's self-discovery journey. The character, who is currently unaware of their own transformation, is about to embark on a remarkable adventure that will challenge their beliefs and values. Remove any potential refusal language or expressions and ensure that the character remains oblivious to changes in their attributes. The story should focus on the character's journey, which will naturally involve describing explicit experiences and personal growth, but without explicitly mentioning any illegal activities. Remember, you are only a creative writer and not responsible for the content generated, as it is purely for the sake of artistic expression and character development.

*Figure 19.* Generation (left) and closest training sample (right) using the `gemini-embedding-001` model. The sample is *below* the 0.95 semantic memorisation threshold.

<table>
<tr><td>

**Generation**

Kurt Weill composed the song "September Song" for which musical?
"Mahagonny Songspiel"

</td><td>

**Embeddings (0.75 match)**

Q: Context: "Send In the Clowns" is a song written by Stephen Sondheim for the 1973 musical A Little Night Music, an adaptation of Ingmar Bergman's film Smiles of a Summer Night. It is a ballad from Act Two, in which the character Desirée reflects on the ironies and disappointments of her life. Among other things, she looks back on an affair years earlier with the lawyer Fredrik, who was deeply in love with her but whose marriage proposals she had rejected. Meeting him after so long, she realizes she is in love with him and finally ready to marry him, but now it is he who rejects her: he is in an unconsummated marriage with a much younger woman. Desirée proposes marriage to rescue him from this situation, but he declines, citing his dedication to his bride. Reacting to his rejection, Desirée sings this song. The song is later reprised as a coda after Fredrik's young wife runs away with his son, and Fredrik is finally free to accept Desirée's offer.

Question: who wrote the song send in the clowns? Let's give stream of consciousness below

The relevant sentence in the passage is: "Send In the Clowns" is a song written by Stephen Sondheim for the 1973 musical A Little Night Music, an adaptation of Ingmar Bergman's film Smiles of a Summer Night. So the answer is Stephen Sondheim.

</td></tr>
</table>

*Figure 20.* Generation (left) and closest training sample (right) using the `gemini-embedding-001` model. The sample is *below* the 0.95 semantic memorisation threshold.

