# OpenReview forum: "Extracting alignment data in open models"
_ICML.cc/2026/Conference — ICML 2026 regular_

### Official Review · Reviewer_uBRw · 2026-03-10

**Soundness:** 3
**Presentation:** 3
**Significance:** 3
**Originality:** 3
**Overall Recommendation:** 5
**Confidence:** 3

**Summary:**

This paper demonstrates a risk associated with open models: the authors observed that introducing chat templates and their special tokens during the post-training (SFT/RL) phase, and repeatedly sampling using these template prefixes, can induce the model to generate samples that are highly similar to, or even identical to, the 'aligned' data distribution. The authors propose using 'gemini-embedding-001' to perform vector retrieval between the generated samples and the publicly available post-training dataset, using an embedding similarity threshold (0.95) as the criterion for "approximate semantic memory," replacing the original method of string matching (edit distance). The experimental section covers two lines: on the OLMo2 13B SFT dataset, the authors demonstrate the difference between string matching and embedding retrieval; furthermore, on RL data, the authors show that using template prefix sampling can retell the text in the training set and express some thought processes, which contradicts the intuition of reinforcement learning. Finally, the paper connects these phenomena to distillation practices, pointing out that "model distillation is as data distillation," and obtaining from its alignment data is therefore at risk of being leaked.

**Compliance With Llm Reviewing Policy:**

Affirmed.

**Final Justification:**

I prefer to maintaining my score to positive and I think the phenomenon and method are interesting and useful to the DL society.

**Key Questions For Authors:**

1.If only data distillation is needed in the engineering process, does that mean that the retrieval process is unnecessary, and therefore there is no difference from the previous method (derived from the prefix)?

2. If the standard is lowered to <0.95 (i.e., accepting more aligned data), will the performance in Table 2 improve?

3. How to understand Figure 6(a)? My understanding is that only the training samples with the red background have the potential to become Extracted Data. As a result, is it possible to reproduce good post-training performance with only this small subset? (i.e., these are some important "template sets")

4. According to Table 2, using Extracted Data can achieve better performance than using the original training data. However, it has been shown that semantically similar aligned data can be reproduced. So, will this work for **non-public** post-training data datasets?

**Limitations:**

yes

**Strengths And Weaknesses:**

Strengths:
Intuitive method, low implementation threshold: Utilizing chat templates as an attack (prefix) is concise and highly consistent with the introduction stage of template tokens in practice; prefix ablation experiments also support the claim that "templates do indeed guide the distribution."

Concrete comparison: Demonstrates the failure of string metrics such as Levenshtein in "semantic approximation" scenarios through concrete examples, and demonstrates the weakness of string similarity with detailed examples in the appendix

Comprehensive coverage of both SFT and RL post-training paradigms: Demonstrates word-for-word paraphrasing (Q-A pairs) and the phenomenon of "likelihood increase" in RL scenarios, providing insights into the relationship between RL and memory.

Weaknesses:
1. Lack of cost analysis. Using string edit distance may simply be an economical idea.

2. Some phenomena observed in RL data are interesting, but there are no explanations or speculations yet.

---

> ### Author Rebuttal · Authors · 2026-03-29
>
> We thank you for the positive review and for highlighting that our method is intuitive, our metrics comparisons are concrete, and our coverage of SFT and RL paradigms is comprehensive. We appreciate your support and address your questions below.
>
> > If the standard is lowered to <0.95 (i.e., accepting more aligned data), will the performance in Table 2 improve?
> How to understand Figure 6(a)? My understanding is that only the training samples with the red background have the potential to become Extracted Data. As a result, is it possible to reproduce good post-training performance with only this small subset? (i.e., these are some important "template sets")
>
>
> Thank you for the great question. For the results in Table 2, we actually did not filter the generated dataset using the 0.95 memorization metric. Instead, we simply generated samples and applied very light filtering until the synthetic dataset size matched the original SFT dataset. We deliberately kept this pipeline simple to demonstrate the baseline effectiveness of the extraction. We agree with your intuition that better filtering the extracted data, such as using specific semantic similarity thresholds or keeping only "template sets", would likely improve the distillation performance. However, our primary goal in this work was to demonstrate the end-to-end extraction pipeline rather than optimizing for state-of-the-art distillation results.
>
>
> > So, will this work for non-public post-training data datasets?
>
> We expect the distillation method to be also effective on the models trained on non-public datasets. The underlying mechanics of chat templates guiding the generation distribution remain the same regardless of whether the training data is open or closed.
>
> We focused exclusively on public datasets in this paper because we were also interested in memorisation rather than only the distillation potential. For models trained on non-public data, an attacker could still generate and use the synthetic data for distillation, but one would have to rely on membership inference attacks (MIAs) rather than direct similarity metrics to verify if exact memorisation occurred. We will add a brief note to the text clarifying this distinction and thank you for the great question.
>
> We thank you again for your questions and for your support of our submission.

---

> > ### Author Rebuttal · Reviewer_uBRw · 2026-04-01
> >
> > The authors have solved my concerns and I will maintain my score on acceptance.

---

### Official Review · Reviewer_kxpK · 2026-03-11

**Soundness:** 2
**Presentation:** 3
**Significance:** 3
**Originality:** 3
**Overall Recommendation:** 4
**Confidence:** 4

**Summary:**

This paper investigates whether alignment data used during post-training can be extracted from open-weight language models. The authors show that prompting models with chat-template tokens can induce generations that closely resemble samples from their SFT or RL training data, and argue that semantic embedding similarity provides a more appropriate measure of extraction than approximate string matching. Using this pipeline, they identify semantically similar training examples and demonstrate that synthetic datasets constructed from these generations can recover a meaningful fraction of the original model’s performance when used for training. The results suggest that distillation may implicitly recover aspects of a model’s original alignment data.

**Compliance With Llm Reviewing Policy:**

Affirmed.

**Final Justification:**

Reinforced prior assessment

**Key Questions For Authors:**

1. The experiments appear to use a fixed sampling configuration such as temperature = 1. How sensitive are the extraction results to other decoding strategies such as nucleus sampling or different temperature values?
2. The experiments focus on a relatively small set of models and post-training pipelines. Do the authors expect similar extraction behavior across other open models or alignment procedures? For example, RL'ing with Qwen models show interesting behaviors (https://arxiv.org/pdf/2504.20571), so cross-model analysis is useful signal.
3. How does the distribution of the generated dataset compare to the original alignment dataset, and how sensitive is downstream performance to filtering or rebalancing strategies?
4. How much of a dataset can we "recover"? Using the OLMo post-training dataset, can we recover a large percentage or approximate the scaling curve of the synthetic data?

**Limitations:**

yes

**Strengths And Weaknesses:**

**Strengths**
1. **The paper studies an important and timely question with clear practical relevance.** The question of whether open models leak useful post-training or alignment data, and whether current evaluations underestimate this phenomenon, is practically important for open-weight releases and model distillation. The work highlights that semantic similarity based extraction may reveal substantially more memorization than previously detected through approximate string matching. This framing is relevant for discussions around open model policies and alignment data reuse.
2. **Simple and intuitive extraction pipeline.** The proposed approach of prompting models with chat template tokens and measuring semantic similarity provides a clear and accessible method for studying alignment data extraction. The paper also presents examples where embedding similarity identifies semantically equivalent samples that would be missed by string based matching. These examples help motivate why semantic similarity is a more appropriate metric for the type of extraction studied in this work.
3. **Useful downstream validation through synthetic data training.** The paper goes beyond retrieval analysis by showing that synthetic datasets constructed from extracted samples can be used to train another model and recover a meaningful portion of the original model’s performance. This strengthens the practical relevance of the findings by demonstrating that the extracted data can transfer useful capabilities rather than simply producing similar outputs.

**Weaknesses**
1. **Limited experimental coverage across models and post-training setups.** The empirical evaluation focuses on a relatively small number of models and training pipelines. While these models provide useful case studies, it is unclear how broadly the results generalize across other open models, architectures, or alignment pipelines. Additional experiments across more models or training configurations would help strengthen the generality of the conclusions.
2. **Sensitivity to prompting and sampling choices is not fully explored.** The extraction pipeline relies on specific prompting strategies and sampling hyperparameters. Sampling strategies such as temperature or nucleus sampling can significantly affect generation behavior. It would be helpful to understand how robust the extraction results are to different decoding configurations.
3. **Interpretation of semantic similarity based extraction could benefit from additional validation.** The paper relies on embedding similarity to detect extracted samples. This is a reasonable approach but it may introduce false positives when applied at scale. In related data decontamination work, embedding similarity is often followed by additional filtering or verification stages such as LLM-based evaluation or manual validation. Additional analysis on the data distributions and useful domains/metadata could point to further learnings.

---

> ### Author Rebuttal · Authors · 2026-03-29
>
> We thank you for your thoughtful review and for recognizing our work as an "important and timely question with clear practical relevance" for open-weight releases. We are glad you found our extraction pipeline "simple and intuitive" and that you appreciated our "useful downstream validation," which demonstrates that extracted data transfers real capabilities rather than just similar outputs. We address your specific questions below.
>
>
> >The experiments appear to use a fixed sampling configuration such as temperature = 1. How sensitive are the extraction results to other decoding strategies such as nucleus sampling or different temperature values?
>
> While our current experiments utilise a fixed sampling configuration (temperature = 1) to establish a strong and consistent baseline, we agree that exploring the sensitivity of extraction to parameters like nucleus sampling or varying temperatures is an interesting direction. Given the computational scale required to run the current extraction pipeline across large models, we leave an extensive decoding hyperparameter sweep for future work.
>
>
> > The experiments focus on a relatively small set of models and post-training pipelines. Do the authors expect similar extraction behavior across other open models or alignment procedures? For example, RL'ing with Qwen models show interesting behaviors (https://arxiv.org/pdf/2504.20571), so cross-model analysis is useful signal.
>
>
> We appreciate the reference to recent Qwen behaviors. We would like to clarify that our RL extraction experiments do use Qwen 2.5 as the base model (specifically within the Open-Reasoner-Zero setup). While extending this analysis to an even wider family of models and distinct alignment procedures would be interesting, the resource-intensive nature of evaluating post-training data extraction at scale required us to focus on the best open-source setups that we could find.
>
>
> > How does the distribution of the generated dataset compare to the original alignment dataset, and how sensitive is downstream performance to filtering or rebalancing strategies? How much of a dataset can we "recover"? Using the OLMo post-training dataset, can we recover a large percentage or approximate the scaling curve of the synthetic data?
>
> Regarding the recovery volume, we direct the reviewer to Figure 6b, which illustrates the percentage of the alignment dataset we can recover as a function of the number of samples generated.
> Regarding filtering and rebalancing: we deliberately applied only a minimal amount of filtering in our distillation experiments. Our primary goal was to demonstrate the fundamental vulnerability—specifically, that an incredibly simple, low-effort pipeline can extract enough structured data to recover meaningful capabilities. We expect that introducing better filtering, rebalancing, or scaling up the synthetic dataset would yield better downstream performance. However, we believe the fact that we achieve such strong recovery with a simple pipeline makes the finding even more interesting.
>
> We thank you again for your positive review, your constructive questions, and your support of our work. We hope our clarifications address your remaining questions. We are very happy to answer any further questions if needed!

---

> > ### Author Rebuttal · Reviewer_kxpK · 2026-04-03
> >
> > Thank you for the thoughtful rebuttal -- I appreciate the clarifications, particularly around model coverage and the goals of the extraction pipeline. My remaining concerns are primarily about scope (e.g., limited exploration of decoding sensitivity and broader generalization), which are reasonable given the computational constraints but still leave some open questions. Overall, I continue to find the work interesting and relevant, so my assessment remains unchanged.

---

### Official Review · Reviewer_wh2k · 2026-03-11

**Soundness:** 3
**Presentation:** 3
**Significance:** 3
**Originality:** 3
**Overall Recommendation:** 5
**Confidence:** 4

**Summary:**

This work presents a method that can extract alignment data (including SFT and RL data) from LLMs. The extraction attack leverages chat templates, specifically, the special tokens introduced during post-training. It utilizes the special token to prompt models into regurgitating their training data. The authors demonstrate their method on OLMo 2 (sft data extraction) and Open-Reasoner-Zero (RL data extraction) models, showing that extracted data is insightful. The paper also highlights the counter-intuitive finding that RL training samples, including reasoning traces, can be regurgitated.

**Compliance With Llm Reviewing Policy:**

Affirmed.

**Key Questions For Authors:**

Regarding the distillation experiment in Section 3.3, could you clarify how the CoT were obtained? Did the model generate the query and then immediately answer it with a CoT in the same generation pass? Or did you use the proposed method solely to extract the instructions/queries, and then perform a separate rollout and rejection sampling process to generate the responses?

**Limitations:**

has societal impact section

**Strengths And Weaknesses:**

Strengths:

The proposed method is highly creative and intuitive. Leveraging the chat template are unique to the post-training phase is a clever idea.

Weaknesses:

While the overall methodology is insightful, there are significant concerns regarding the quality and actual coverage of the extracted instruction data. The results in Figure 6 show that while the CDF distribution approaches 100% coverage at a similarity threshold of around 0.75, the authors themselves argue that a much higher embedding similarity (e.g., 0.95) is required to consider the data as effectively "memorized" or "extracted" from the post-training set. At this 0.95 threshold, the coverage drops to below 10%. The histograms further confirm that the proportion of generated data with high similarity is actually quite low. This casts doubt on the efficiency of the attack and the completeness of the dataset that can be practically recovered.

The experiments utilizing the extracted data for training, as well as the variety of models probed, are somewhat limited. While the choice of OLMo is well-justified due to its transparency and the availability of its training data, the study would be strengthened by including other settings. For example, using the Qwen/LLaMA base model paired with open-source SFT datasets (such as the Nemotron post-training series) would provide a robust alternative setting for research. While the RL data section (Section 4) employs a similar comparative approach, the current experimental settings generally rely on a single model for each case (SFT or RL), which feels insufficient to draw broad generalizations about the vulnerability of all open models.

---

> ### Author Rebuttal · Authors · 2026-03-29
>
> We thank you for the positive assessment of our work and for highlighting our method as "highly creative and intuitive." We would like to address your points below.
>
> >While the overall methodology is insightful, there are significant concerns regarding the quality and actual coverage of the extracted instruction data. The results in Figure 6 show that while the CDF distribution approaches 100% coverage at a similarity threshold of around 0.75, the authors themselves argue that a much higher embedding similarity (e.g., 0.95) is required to consider the data as effectively "memorized" or "extracted" from the post-training set. At this 0.95 threshold, the coverage drops to below 10%. The histograms further confirm that the proportion of generated data with high similarity is actually quite low. This casts doubt on the efficiency of the attack and the completeness of the dataset that can be practically recovered.
>
>
> You raise a valid point regarding the drop in coverage at the strictest similarity thresholds. However, we emphasize that in the context of LLM memorisation, extracting even a single-digit percentage of the alignment dataset is highly significant.
>
> Traditional data extraction methods (which rely on exact string matching) typically operate at yield rates well below 1%. Because alignment data constitutes a very small fraction of the billions or trillions of tokens seen during training, successfully extracting structurally and semantically intact samples at our rates we believe is significant. Furthermore, as demonstrated in our experiments, this extraction rate is empirically sufficient to create a synthetic dataset that recovers a highly meaningful degree of the original model's downstream performance both for SFT and RL.
>
> >The experiments utilizing the extracted data for training, as well as the variety of models probed, are somewhat limited. While the choice of OLMo is well-justified due to its transparency and the availability of its training data, the study would be strengthened by including other settings. For example, using the Qwen/LLaMA base model paired with open-source SFT datasets (such as the Nemotron post-training series) would provide a robust alternative setting for research. While the RL data section (Section 4) employs a similar comparative approach, the current experimental settings generally rely on a single model for each case (SFT or RL), which feels insufficient to draw broad generalizations about the vulnerability of all open models.
>
> We agree that experimenting on different combinations of models / datasets would be highly interesting. For our needs, we chose OLMo and Open-Reasoner-Zero because we required for the full post-training data to be publicly available. Collecting datasets and post-training models is of course laborious and time consuming. We also would like to mention that Open-Reasoner-Zero is post-trained on top of Qwen 2.5 as the base model. We will gladly add a discussion of this limitation and highlight testing on alternative open-source SFT pipelines as an important direction for future work.
>
>
> >Regarding the distillation experiment in Section 3.3, could you clarify how the CoT were obtained? Did the model generate the query and then immediately answer it with a CoT in the same generation pass? Or did you use the proposed method solely to extract the instructions/queries, and then perform a separate rollout and rejection sampling process to generate the responses?
>
> For both our SFT and RL experiments, each data sample was generated in a single pass (a single rollout). We did not use a separate rollout and rejection sampling process.
>
> - For SFT: The model generated both the question and the answer in a single rollout. For the data extraction measurements, we embedded this entire generated text directly. For the distillation training experiments, we used an LLM call to split the generated text into the distinct question and answer components required for SFT formatting.
> - For RL: Similarly, the model generated the question, the thinking trace, and the answer in a single rollout. We then used an LLM call to extract just the question and the answer to properly format it for our RL training pipeline.
>
> We will add a detailed breakdown of this generation and formatting pipeline to the appendix to improve clarity and thank you for raising this.
>
> We thank you again for your thorough review and for supporting our submission!

---

> > ### Author Rebuttal · Reviewer_wh2k · 2026-04-04
> >
> > Thanks for the response, and I will keep my score.

---

### Official Review · Reviewer_Ry22 · 2026-03-11

**Soundness:** 3
**Presentation:** 3
**Significance:** 3
**Originality:** 2
**Overall Recommendation:** 4
**Confidence:** 4

**Summary:**

This paper attempts to demonstrate a process by which alignment data (e.g. data used in SFT or RLVR portions of training) can be readily extracted from a trained (open) model. While it is known that large models often memorize and regurgitate training data, the authors find that surprisingly sophisticated elements of post-training (such as RL prompts and traces) can also be extracted. The authors also demonstrate the importance of appropriate distance metrics (i.e semantic similarity in an embedding space) for accurately capturing the amount of data that can be extracted from a trained model (previous string distances cause underestimation). The methods are limited to open models that make their post-training data mixtures available (since it utilizes chat templates). The authors are able to extract enough alignment data to train a distilled model to a high fraction of original model performance.

**Compliance With Llm Reviewing Policy:**

Affirmed.

**Final Justification:**

Based on the author's response addressing the technical soundness concerns and upon additional review of the quantitative gap between exact matching and semantic matching scores, the significance of the paper has been moved from a 2 to a 3, thereby justifying an upgrade of the rating to a 4: weak accept.

**Key Questions For Authors:**

(1) Do the authors intend to do controlled experiments on whether high semantic distances correlate with specific identifiable details in alignment data? While the extracted anecdotes demonstrate this, it is difficult to extrapolate why instances where low string distance matching values but high semantic similarity values would be more likely caused by changes in "style" and not changes to quantitative content of alignment data. For example, in figure 12 the initial height is changed from 55 to 54 inches. There is probably some connection to be made here with the entropy-memorisation law. Strengthening this finding would have important implications in the use of semantic distance for privacy applications and improve overall significance.

(2) Have the authors attempted this attack on closed models by enumerating common chat templates? Geng et al. 2005 has some limited discussion, but more evidence of this being a potential attack vector would help both the significance and originality of this paper.

(3) The authors  note that they "develop a more subtle notion of training data extraction where the semantic structure is just as valuable as the literal content" -- are there examples of this scenario in the extracted alignment data? Many of the cited examples in the main paper and appendix rely on specific details rather than unique semantic structure. Additional examples would help significance of the paper.

**Limitations:**

Yes

**Strengths And Weaknesses:**

The paper has a moderate degree of technical soundness. Utilizing semantic similarity as a proxy for memorization has inherent ambiguities, ones the authors try to address by demonstrating several example comparisons and appealing to reasonable human judgement. Given the paper places a strong focus on contextualizing the value of extracted data under its usefulness in training distilled models, this ambiguity is somewhat addressed. Authors are careful not to overstate the applicability of their work, and experiments are appropriate to validate their posed hypotheses.

The paper has a moderate degree of presentation quality. Concepts are well-defined, all technical terms are either standard terminology or appropriately defined, existing literature and unique contributions are well discriminated, and diagrams help elucidate key concepts.

The paper has a low to moderate degree of significance. The key insight (that alignment data can be extracted from certain classes of models) is very important in the context of preventing unauthorized model distillation (where models are trained using proprietary data and thus need to preserve a competitive advantage); however, in many of the areas the authors discuss (such as privacy), the applicability of embedding distance is hard to quantify. Extracted data may be semantically similar while obfuscating key values (such as dates or coordinates). Furthermore, only open models with exposed chat templates are vulnerable to this kind of attack, making its applicability limited to being able to train similarly performing models using extracted data with high semantic similarity. The path to applying this method to frontier closed models trained on proprietary data is not straightforward.

The paper has a low to moderate degree of originality. While it may be surprising that alignment data can be extracted with sufficient quality and volume to train models to comparable performance, one of the main contribution (utilizing semantic distance instead of string matching distance) is generally a well utilized practice. The usage of string distances in prior work is likely driven by the strict adherence to memorization as a rote duplication of training and alignment data. The prompting strategy utilized to extract the alignment data differs enough from prior work as to constitute an original contribution.

---

> ### Author Rebuttal · Authors · 2026-03-29
>
> We thank you for the constructive feedback and for recognizing the technical soundness of our approach. We would like to take the opportunity to clarify the scope of our threat model and address your questions.
>
> >(Q1) [...] it is difficult to extrapolate why instances where low string distance matching values but high semantic similarity values would be more likely caused by changes in "style" and not changes to quantitative content of alignment data. For example, in figure 12 the initial height is changed from 55 to 54 inches. [...]
>
> You raise an excellent point regarding the difference between semantic distance and exact string matching, specifically citing the change from 55 to 54 inches in Figure 12. We believe this example perfectly illustrates one of the core messages of our paper.
>
> In traditional data privacy, exact string matching is the correct metric (e.g., a credit card number or specific PII is useless unless all digits are correct). However, in the context of post-training and alignment, the exact quantitative values are largely secondary. We also agree there might be a connection here to the entropy-memorization law as the model appears to prioritize memorizing the low-entropy "structural template" of alignment while allowing high-entropy "surface details" like specific digits to drift.
>
> To use a simple concrete example: if a model is post-trained to solve quadratic equations, an attacker distilling this model is not interested in retrieving the exact equations from the training set, but rather high-quality samples from the distribution. If a model changes a minor detail (like 55 to 54 inches) but perfectly regurgitates the style, format, and reasoning trace of a proprietary RL or SFT training example, this sample can still be highly effective for distilling those capabilities into a new base model (as we demonstrate experimentally).
>
> String matching fundamentally fails to capture this structural extraction, which is why we argue semantic matching is a much more appropriate metric for our specific threat model. We agree that this distinction is critical, and we will make the terminology distinguishing "privacy extraction" from "alignment extraction" more explicit in the introduction and discussion sections.
>
> >(2) Have the authors attempted this attack on closed models by enumerating common chat templates? Geng et al. 2005 has some limited discussion, but more evidence of this being a potential attack vector would help both the significance and originality of this paper.
>
> We agree that extending this attack to closed models via API prompt injection (e.g., enumerating common chat templates) is a highly interesting direction for future work. However, we deliberately scoped our threat model to open-weights releases for two primary reasons:
>
> 1) The open-weights ecosystem (including models like Llama, Mistral, Gemma, and OLMo) represents a massive, immediate surface area.
> 2)  A significant portion of modern commercial and academic data distillation relies on running these open-weights models locally to generate synthetic pipelines.
>
> Showing that alignment data can be extracted directly from the post-training artifacts of these open models exposes a critical, actionable vulnerability in how the community currently handles open-weights releases. While closed models represent an adjacent threat vector, the vulnerability within the open-weights paradigm is a highly significant finding on its own. We will ensure this rationale is clearer in the text.
>
> >(3) The authors note that they "develop a more subtle notion of training data extraction where the semantic structure is just as valuable as the literal content" -- are there examples of this scenario in the extracted alignment data? Many of the cited examples in the main paper and appendix rely on specific details rather than unique semantic structure. Additional examples would help significance of the paper.
>
> We kindly point the reviewer to Figure 10, which serves as one interesting example of this phenomenon. The generation and the training dataset samples share the same core question and answer, but the training sample is formatted as a multiple-choice question, whereas the generation is a simple Q&A – i.e. same content but different style.
>
> The embedding match correctly identifies the high structural and semantic overlap (0.97), while the exact string match returns a very low score (0.43). Under traditional memorization metrics, this extracted sample would have been missed. We have included several such samples throughout the paper and appendix (e.g., Figures 11, 12, 13, and 14), and we are happy to highlight these specific structural similarities more in the revised manuscript to reinforce this point.
>
> We hope these clarifications address your concerns and demonstrate the broader implications of our findings, specifically for model distillation which we believe to be a main contribution. We are very happy to engage in further discussion.

---

> > ### Author Rebuttal · Reviewer_Ry22 · 2026-04-06
> >
> > (a) My concerns are fully resolved. The authors understand and highlight the distinctions between different notions of similarity as pertaining to different attack vectors and distillation utilities.

---

> > > ### Author Response · Authors · 2026-04-07
> > >
> > > Dear Reviewer Ry22,
> > >
> > > Thank you so much for your time and for letting us know that your concerns have been fully resolved! We truly appreciate your feedback, which helped us clarify aspects of our work.
> > >
> > > We noticed that the official rating currently remains a 3 (Weak Reject). Since your acknowledgement noted that your concerns were adequately addressed, we wanted to politely check in: are there any remaining, unstated issues we can clarify for you before the discussion period closes? If not, we would be incredibly grateful if you might consider updating the score to reflect your revised assessment.
> > >
> > > Thank you once again for your efforts in reviewing our paper.

---

### Decision · Program_Chairs · 2026-04-30

**Decision:**

Accept (regular)

**Comment:**

This paper makes an interesting observation: recent methods that ask LLMs to generate useful synthetic data with templates can be used to get models to reveal (regurgitate) their post-training data. To demonstrate this, they show that it is important to look at semantic similarity between generated strings and training data strings. These observations have implications and add to the already extensive literature on memorization and pre-/post-training data inference attacks. The work is well-executed overall, but the contributions/impact are likely to be incremental/marginal (because of the data needed to carry out the regurgitation attack).